# Pandemic-Induced Reductions on Swim Training Volume and Performance in Collegiate Swimmers

**DOI:** 10.3390/ijerph19010164

**Published:** 2021-12-24

**Authors:** Gloria Martinez Perez, Matthew VanSumeren, Michael Brown, Tamara Hew-Butler

**Affiliations:** Kinesiology, Health, and Sport Studies, Wayne State University, Detroit, MI 48202, USA; gloria.martinez@wayne.edu (G.M.P.); msvansum@wayne.edu (M.V.); mbrown17@wayne.edu (M.B.)

**Keywords:** swimming, COVID-19, swim performance

## Abstract

The COVID-19 pandemic caused significant training disruptions during the 2020–2021 season, due to lockdowns, quarantines, and strict adherence to the pandemic protocols. The main purpose of this study was to determine how the pandemic training restrictions affected training volume and performance in one collegiate swim team. Cumulative training volume data across a 28-week season were compared between a pandemic (2020–2021) versus non-pandemic (2019–2020) season. The swimmers were categorized into three groups (sprinters, mid-distance, and long-distance) based on their training group. The performance times of 25 swimmers who competed in the regional championships, during both the non-pandemic and pandemic year, were compared via one-way ANOVA. Twenty-six male and 22 female swimmers commenced the 2020–2021 (pandemic) season, with 23% of the swimmers voluntarily opting out. Three COVID-19 cases were confirmed (2%) by the medical staff, with no long-term effects. Significant reductions in the average swim volume were verified in sprinters (32,867 ± 10,135 vs. 14,800 ± 7995 yards; *p* < 0.001), mid-distance (26,457 ± 10,692 vs. 17,054 ± 9.923 yards; *p* < 0.001), and long-distance (37,600 ± 14,430 vs. 22,254 ± 14,418 yards; *p* < 0.001) swimmers (non-pandemic vs. pandemic season, respectively). In the regional performance analyses, the sprinters swam faster (*n* = 8; −0.5 ± 0.6 s), while the mid-distance (*n* = 10; 0.17 ± 2.1 s) and long-distance (*n* = 7; 6.0 ± 4.9 s) swimmers swam slower (F = 11.76; *p* = 0.0003; r^2^ = 0.52). Thus, the pandemic caused significant reductions in swim training volume, with sprinters performing better and long-distance swimmers performing worse at the regional championships.

## 1. Introduction

For collegiate athletes, the coronavirus disease 2019 (COVID-19) pandemic altered their training and competition [1,2]. Team outbreaks [3], unplanned lockdowns [2], frequent quarantines [4], and delays associated with the routine testing, masking, and social distance protocols [4,5] severely undermined both training quality and quantity. Moreover, the athlete-specific health risks associated with competitive exercise included the development of myocarditis [6], increased transmission with close-contact sports [7,8], and/or augmented infection risk due to excessive exercise-induced immunosuppression [9]. However, these hypothesized risks to athletes remain complex and controversial [7,10,11]. 

The current data on the effect of pandemic-induced restrictions on athlete health, swim training, and performance remains sparse. The cancellation of the U.S. National Collegiate Athletic Association (NCAA) championships, one of the most major Fédration Internationale de Natation (FINA) swimming events [2], and the postponement of the Tokyo Olympics [12] in 2020 forced elite swimmers to train alone and often at home [2]. Serendipitously, the effect of forced reductions in swim training volume on performance indirectly provided insight into the quality versus quantity debate, which suggests that elite-level swimmers may not require high swim volumes to perform well [13,14,15,16].

Thus, the primary aims of this retrospective, observational study were to (1) determine how the pandemic-induced training restrictions affected training volume in a NCAA division 2 (NCAA D2) swim team (compared with a non-pandemic year), and (2) evaluate the differences in swim performance on swimmers who competed at the NCAA D2 regional championship meet during a pandemic (2019–2020) versus a non-pandemic (2020–2021) year. A secondary aim of this study compared the top 16 swimming performances for all swimming events held at both the regional and national NCAA D2 meets during a pandemic versus a non-pandemic season. We hypothesized that the effect of pandemic-induced restrictions on training volume would result in decreased swimming performances at all levels of competition (i.e., at the local, regional, and national competitive levels).

## 2. Materials and Methods

### 2.1. Data from a Single Local Swim Team

For the primary aims of this study, observational and retrospective data were collected from the official training logs obtained from the swimming coaches from a single NCAA D2 swim team, at a midwestern university located within the United States (U.S.). Cumulative and average training volume data across a 28-week season were compared between a pandemic (2020–2021) and a non-pandemic (2019–2020) competitive season. We report training volume (i.e., swimming distance) in yards (yd) as the NCAA championship events are conducted and reported in yards, rather than meters. Swim training volume data were de-identified prior to analysis, and the study protocol was reviewed by Wayne State University’s IRB and found to qualify for an exemption according to category 4 (IRB-21-10-4075-B3 Expedited/Exempt-EXEMPT, approval date 14 October 2021).

Forty-eight (26 males) swimmers started the 2020–2021 (pandemic) season as official student athletes on our (“local”) collegiate swim team. These local swim team data were further sub-divided into the following three groups (males and females combined): sprinters (50 yd–100 yd), mid-distance (200 yd–400 yd), and long-distance (500 yd–1650 yd) swimmers, according to their assigned training groups for each season. The only demographic variable available to us was self-reported age at the commencement of the pandemic year. Swim training volumes were analyzed as a cumulative average (28 weeks, from September to March, with a one-week hiatus for the winter/Christmas break before week 17) between the pandemic (2020–2021) and non-pandemic (2019–2020) year. Swim training volume was compared during the pandemic versus non-pandemic years using 2-way ANOVA (swim training group as the column factor and pandemic versus non-pandemic year as the row factor).

To assess swimming performance, the finishing times (reported in seconds) for 25 local university swimmers who competed in the Great Lakes Intercollegiate Athletics Conference (GLIAC) Regional Championships, during both the pandemic and non-pandemic years, were evaluated. Swim performance data were analyzed as the change (∆) value between the pandemic and non-pandemic swim performance time. We compared performance times by training group using 1-way ANOVA. 

### 2.2. Data Obtained from Regional and National Championship Meets

For the secondary aims of this study, we compared the (publicly available) top 16 swimming performance times, for all events, achieved at both the GLIAC Regional [17,18] and NCAA D2 [19,20] Championships during the pandemic (2020–2021) versus non-pandemic years (2019–2020 for the GLIAC and 2018–2019 for the NCAA meet, as NCAA Championships were cancelled in 2020). The competition format for all events consisted of two heats plus the event final [21]. We chose the top 16 times (from the two heats) of all swimmers who made it to the final event (i.e., the top qualifying times required to make the final event) as our standard convention. Non-paired t-tests were used to evaluate statistically significant differences in performance time, per event, between the pandemic and closest non-pandemic year. 

All data are reported as means ± SD, with statistical significance set a priori at *p* < 0.05.

## 3. Results

The university remained fully online from March 2020 to September 2021. However, select sports teams (including swimming) were allowed to return to campus in September 2020 (fall semester) to commence training for their competition, after a four-month summer break (May to August). The pandemic-induced restrictions for the swim team included frequent (3x/week) COVID-19 testing via nasopharynx swabs, which tested for the severe acute respiratory syndrome coronavirus 2 (SARS-CoV-2) virus via polymerase chain reaction (PCR), for all swimmers and coaches participating in the competitive season. Swimmers with a positive test, or those who were exposed to COVID-19, were required to quarantine for 10 days. Due to social distancing protocols, only eight swimmers were allowed in the pool at one time (unlimited swimmers were allowed in the pool during non-pandemic times). 

Overall, 26 male (average age: 20.2 years, range: 18–24) and 22 female (average age: 19.9 years, range: 18–22) swimmers started the 2020–2021 (pandemic) season, with six males (23%) and five females (23%) voluntarily opting out by the end of the season. During a typical non-pandemic year (such as 2019–2020, which included 26 males and 17 females), no swimmers voluntarily dropped out. A total of three COVID-19 cases were confirmed (2%) by the medical staff during the 28-week pandemic season, with no long-term effects on athlete health or performance following COVID-19 recovery. The three confirmed COVID-19 cases were associated with mild (1) or no (2) symptoms. The return-to-play protocol consisted of a 10-day quarantine followed by a monitored, gradual return to play [22]. 

Figure 1 illustrates the weekly swimming distance, plus the completed and cancelled competitions, in the non-pandemic (2019–2020) and pandemic (2020–2021) years. Of note, there is a one-week training hiatus for final exams (prior to week 17), and three taper periods before the mid-season competition (Denison Invite, cancelled in the pandemic year), GLIAC Regional, and NCAA Championship competitions.

### 3.1. Swim Training and Performance Results from a Single Local Swim Team

There were significant reductions in the average weekly swim volume in sprinters (average age: 20.3 years), mid-distance (average age: 19.9 years), and long-distance (average age: 20.2 years) swimmers (non-pandemic vs. pandemic season, respectively) during the 28-week training period (Figure 2). 

The cumulative training distance was reduced (non-pandemic vs. pandemic year) in sprinters (668,275 vs. 414,400 yd; 62%), mid-distance (740,800 vs. 477,500 yd; 64%), and long-distance (1,052,800 vs. 623,100 yd; 59%) swimmers.

For the 25 swimmers who competed in the regional championships, in both the pandemic and non-pandemic years, the sprinters swam faster (*n* = 8; −0.5 ± 0.6 s), while the mid-distance (*n* = 10; 0.17 ± 2.1 s) and long-distance (*n* = 7; 6.0 ± 4.9 s) swimmers swam slower (F = 11.76; *p* = 0.0003; r^2^ = 0.52) during the pandemic year compared with the previous non-pandemic year. Of note, improved performance times occurred for the sprinters, despite the significant reductions in training volume (Figure 3).

### 3.2. Performance Results from the Regional and National Championship Meets

In the GLIAC Regional Swim Championship meet, the female swimmers swam significantly slower (as a mean value) in the 200 yd freestyle (FR), 500 yd FR, 200 yd butterfly (FL), 100 yd backstroke (BK), and 200 yd individual medley (IM) in the pandemic (2020–2021) versus non-pandemic (2019–2020) year (Table 1).

In the GLIAC Regional Swim Championship meet, the male swimmers swam significantly slower (as a mean value) in the 500 yd FR in the pandemic (2020–2021) versus non-pandemic (2019–2020) year (Table 2).

In the NCAA D2 National Swim Championship meet, the female swimmers swam significantly slower (as a mean value) in the 50 yd FR and 100 yd FR, while they swam significantly faster in the 400 yd IM in the pandemic (2020–2021) versus non-pandemic (2018–2019) year (Table 3).

In the NCAA D2 National Swim Championship meet, the male swimmers swam significantly slower (as a mean value) in the 200 yd BK, while they swam significantly faster in the 50 yd FR in the pandemic (2020–2021) versus non-pandemic (2018–2019) year (Table 4).

## 4. Discussion

From the overarching perspective of sports safety, the incidence of COVID-19 was low (2%) in our cohort of NCAA D2 swimmers during the 2020–2021 season. Two of the three COVID-19 positive swimmers were asymptomatic (67%). This finding is in agreement with a previous study conducted on a large (10,265 students) collegiate population, which demonstrated that >50% of positive COVID-19 cases were without symptoms [23]. Fortunately, all three of our COVID-19 swimmers returned to training and competition without any lingering side effects. This finding agrees with a previous study conducted on 46 elite Hungarian swimmers, where all 14 (30%) COVID-19 positive swimmers recovered and resumed training for the Tokyo Olympics [12]. Additionally, only 1.2% of 3529 COVID-19 positive NCAA student athletes reported lingering symptomatology >3 weeks following SARS-CoV-2 infection [24]. Our rigorous COVID-19 testing protocol (COVID-19 testing 3x/week) appeared to be effective in reducing the spread of COVID-19 amongst athletes, as similarly verified in a cohort of Southeastern Conference collegiate football players [7]. Most COVID-19 spread amongst athletes seems to occur at social gatherings and in communal living arrangements, rather than during official training and competition, [4,7] as anecdotally supported in our swimming cohort. 

From a mental health perspective, ~1 in 4 (23%) of our swimmers opted (dropped) out of the season during the pandemic year. This drop-out rate was most likely due to enhanced psychological stress, rather than training distress from the pandemic [25,26]. A recent (pre-pandemic) systematic review conducted on swimmers supports this assumption, citing the common reasons for dropping out as “pressure” and “lack of fun” [27]. Of note, the most common reasons for dropping out of swimming appear to be within an athlete’s control [27]. This suggests that in future pandemics, strategies to maximize an athlete’s sense of “control” (through freely available mental health counseling) should be prioritized to minimize student athlete drop-out rates. 

Our drop-out rate was equal between the sexes. Of note, mental health symptoms are typically higher in female versus male elite athletes [27,28]. However, males are more prone to anger [29] and are more reluctant to seek help for mental health problems [30] than females. Thus, differential coping strategies may help explain the equal drop-out rate between male and female swimmers in our local cohort.

### 4.1. Swim Training and Performance Results from a Single Local Swim Team

The pandemic-induced restrictions resulted in significant reductions in overall swim training volumes in our collegiate swim team, as expected from the COVID-19 social distancing protocols, lockdowns, and quarantines [2]. Despite the cumulative decrease in swim training volume during the pandemic, the sprinters performed better (i.e., swam faster), while the mid-distance swimmers performed similarly. The long-distance swimmers, however, performed worse (i.e., swam slower). This performance decline in the long-distance swimmers was expected, as a previous report conducted on 18 professional cyclists documented a 33.9% decrease in total training volume, which resulted in a 1% and 19% decrease in 5-min and 20-min time-trial performance time, respectively [1]. Similarly, a typical long-distance swim race lasts between ~4 and ~15 min (Table 1, Table 2, Table 3 and Table 4).

Anecdotally, because of the limited time spent in the pool during the pandemic, all swim workouts were intended to be performed at higher intensities. We cannot quantify this observation with any valid metrics, but this anecdotal assumption would favor training adaptations in sprinters, but not in long-distance swimmers. Additionally, the same land workouts were prescribed by the coach, and mainly consisted of body circuit exercises at home, which were not supervised. Because of the young age of these collegiate swimmers (~20 years), age-related declines in performance were not expected to contribute to the performance differences observed between the sprinters and distance swimmers, since performance in swimmers peaks at around the age of 21 years [31,32]. 

Our finding that sprinters performed better with reduced swim training volumes also indirectly informs a larger debate regarding the efficacy of quality versus quantity training on swimmers’ performance [13,14,15,16] and health [33]. A growing body of evidence suggests that a high swim training volume does not offer performance advantages over a low swim training volume past adolescence [34]. Furthermore, since 81% of the 32 Olympic pool events cover at least, or less than, 200 m in total distance, with an average race duration of less than 140 s, the efficacy of high swimming volumes (aerobic training) on anaerobic sprinting performance appears to be maladaptive [13,14,15] in lieu of shorter race-paced training [16]. With additional regard to the health risks, high swim training volumes have also been associated with shoulder pain in adolescent swimmers [33], as well as overtraining and burnout [35]. Thus, our results support the training concept that low swim volume training may offer performance advantages to sprinters, but is likely to be deleterious to long-distance swimming performance. 

### 4.2. Performance Results from the Regional and National Championship Meets

The largest pandemic-induced decreases in average (top 16) performance times were noted in the female swimmers competing at the GLIAC Regional Championships, where 36% of the races (5 of 14) recorded significantly slower average racing times during the pandemic compared to the non-pandemic year. For the male swimmers competing in the GLIAC Championship meet, only one race (7%), the 500 yd FR, recorded a slower average finishing time during the pandemic (versus non-pandemic) year. 

The reasons why females swam slower in the GLIAC Championships during the pandemic year remain unclear and multifactorial. We can only speculate that females—especially at lower levels of competition (i.e., at the D2 level, where academics and athletics are more “balanced” [36])—are historically more susceptible to mental health challenges [27,28], burnout [35], and/or overtraining [37], which may affect their performance. However, this regional finding conflicts with our local findings on drop-out rates, which were equal between males and females. Thus, the conflicting sex differences regarding performance and drop-out rates requires further investigation at the local, regional, and national levels.

At the NCAA D2 National Championship meet, however, the top finishing times (on average) were not as adversely affected. In some events, such as the 400 m IM for females and the 50 FR for males, the swimmers unexpectedly performed faster (on average) during the pandemic year. This finding that the pandemic-induced restrictions were not deleterious to swim performances at the national championship level is supported by previous data suggesting that short periods of COVID-19-induced detraining did not adversely affect performance in the Hungarian swimmers training for the Tokyo Olympics [12]. Thus, the performance effects of the COVID-19 lockdown are higher at the local level than at the elite national level for reasons that require further clarification.

### 4.3. Practical Implications

The practical points obtained from a critical analysis of retrospective and observational data suggest the following: (1) almost one-quarter (~23%) of collegiate NCAA D2 swimmers are likely to drop out from official team activities without added interventions (such as mental health support); (2) thrice weekly COVID-19 testing appears to limit the spread of COVID-19 amongst competitive swimmers; (3) COVID-19 positive student athletes do not appear to suffer long-term health or performance consequences; (4) swim training volumes are significantly reduced during a pandemic, which negatively impacts long-distance swimming performance; (5) sprinters appear to perform better with reduced swim training volumes, which supports the quality over quantity debate; (6) swim performance is least impacted at the highest level of collegiate competition (NCAA Championships). Thus, as Michigan currently faces another winter COVID-19 surge, these data serve to inform strategies that reduce the deleterious consequences of a pandemic on student athlete health, training, and swim performance. 

### 4.4. Strengths and Limitations

The strengths of these data are that swimming volumes and performance times are routinely recorded and easily reproduceable. The potential confounding effects of the external environment—such as ambient temperature, wind speed, and variable terrain—are relatively controlled at indoor swimming venues, regardless of the location of the pool. As such, comparing swim performance times across both time (year) and space (location) represent reliable comparisons (unlike outdoor sporting events). 

The limitations of these data include our reliance on existing records as a primary source of data, and the inability to quantify the amount of dry-land training the swimmers received. Previous studies suggest that dry-land training may enhance swimming performance in both pandemic [38] and non-pandemic [39] times, which may have confounded our swim performance results. Furthermore, we were unable to quantify the training load during the off-season (May to September), nor quantify the training intensity (because human subject research was prohibited for 16 months during the pandemic). Lastly, we duly acknowledge the limitations of comparing different cohorts of swimmers in the GLIAC Regional and NCAA Championships in different years. Comparing the same swimmers across time would have been ideal, but was not practical beyond the local level.

## 5. Conclusions

The reduced, pandemic-induced swim training volumes positively impacted sprinting performance, while negatively impacting long-distance swim performance in a cohort of midwestern NCAA D2 swimmers. Swim performance declines during the pandemic were mostly evident in female swimmers competing at regional championship events. The average swim performance was largely unaffected at the national level (in a pandemic versus non-pandemic year). While a substantial (23%) number of swimmers dropped out during the pandemic season, our overall COVID-19 transmission rates were low, with all (2%) COVID-19 positive swimmers recovering without ill effects.

## Figures and Tables

**Figure 1 ijerph-19-00164-f001:**
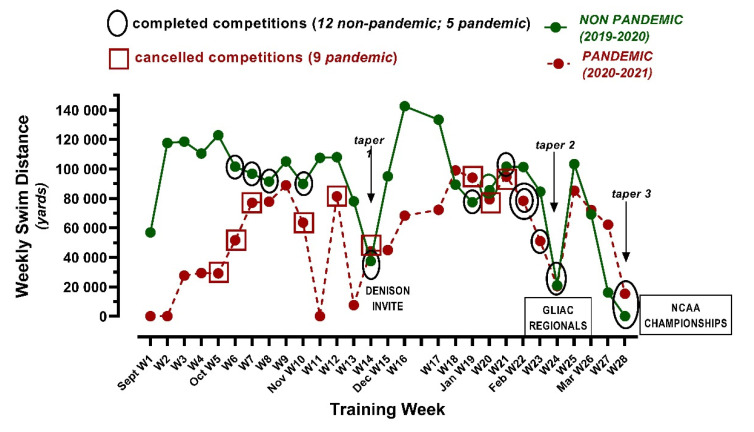
This schematic summarizes the weekly swimming distance/volume completed for swimmers during the non-pandemic (2019–2020) and pandemic (2020–2021) seasons. All scheduled competitions that were completed (circles) or cancelled (squares) are denoted. Three “taper” periods represent planned reductions in training distance/volume prior to the following three major competitions: the mid-season Denison Invite (cancelled during the pandemic year), the GLIAC Regional Championships, and NCAA Championships.

**Figure 2 ijerph-19-00164-f002:**
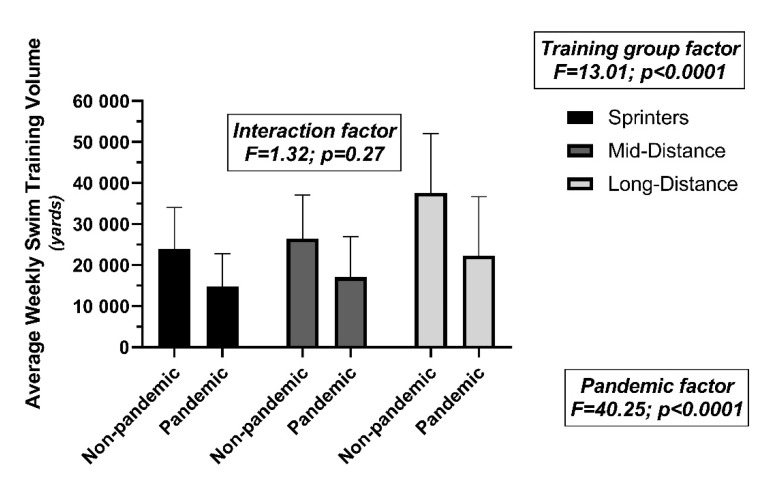
Comparisons of average weekly training swim volumes for sprinters, mid-distance, and long-distance swimmers (training group) during both the non-pandemic and pandemic (pandemic factor) years.

**Figure 3 ijerph-19-00164-f003:**
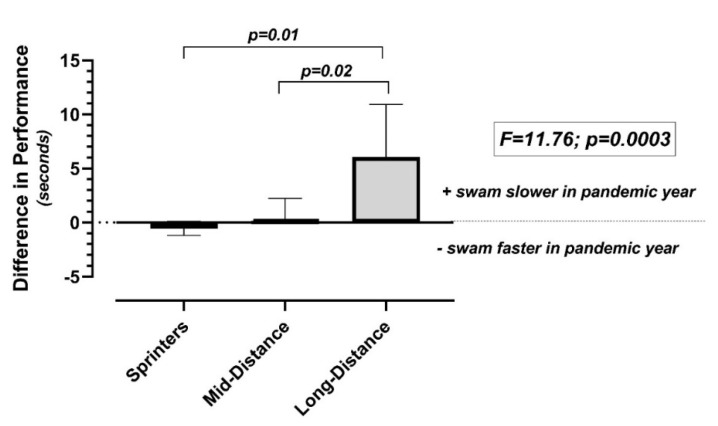
Comparison of performance in the GLIAC Regional Championship meet in swimmers competing in both the pandemic and non-pandemic years, categorized by training group.

**Table 1 ijerph-19-00164-t001:** Top 16 female swimming performances in the GLIAC Regional Championship meets, in a non-pandemic versus pandemic year. All swim events are reported in yards (yd) while all finishing times are reported in seconds.

Swim Event	Performance Times	Performance Times
	Non-Pandemic	Pandemic
	(2019–2020)	(2020–2021)
	Mean ± SD	Mean ± SD
	*(Fastest–Slowest)*	*(Fastest–Slowest)*
50 FR	24.3 ± 0.3	24.3 ± 0.4
	*(23.8–24.8)*	*(23.6–24.9)*
100 FR	52.9 ± 0.9	52.8 ± 0.5
	*(51.4–53.8)*	*(51.9–53.5)*
** *200 FR* **	** ** 114.4 ± 1.2* **	** *115.5 ± 1.6* **
	** *(111.3–115.7)* **	** *(112.9–117.5)* **
** *500 FR* **	** ** 307.3 ± 5.1* **	** *311.2 ± 5.7* **
	** *(298.5–315.9)* **	** *(298.5–318.1)* **
1000 FR	627.1 ± 13.2	636.3 ± 14.7
	*(605.6–649.2)*	*(603.8–654.3)*
1650 FR	1111.3 ± 228.1	1080.9 ± 40.9
	*(1011.6–1961.4)*	*(1013.4–1165.7)*
100 FL	57.7 ± 1.23	58.4 ± 1.4
	*(55.5–59.1)*	*(56.8–60.5)*
** *200 FL* **	** ** 130 ± 3.4* **	** *135.1 ± 7.7* **
	** *(123.4–134.5)* **	** *(123.3–154.2)* **
** *100 BK* **	** ** 57.7 ± 1.0* **	** *58.7 ± 1.4* **
	** *(55.5–58.8)* **	** *(55.7–61.0)* **
200 BK	126.5 ± 4.1	129.3 ± 4.3
	*(119.7–132.7)*	*(119.9–136.1)*
100 BR	66.0 ± 1.3	66.4 ± 1.6
	*(63.6–68.0)*	*(63.1–70.3)*
200 BR	145.4 ± 4.2	146.4 ± 3.9
	*(139.3–151.3)*	*(139.5–152.1)*
** *200 IM* **	** *** 128.0 ± 2.1* **	** *130.1 ± 2.4* **
	** *(123.7–130.4)* **	** *(125.7–133.6)* **
400 IM	276.9 ± 6.6	284.8 ± 16.5
	*(268.4–290.5)*	*(269.2–314.4)*

* *p* < 0.05; ** *p* < 0.01 between non-pandemic and pandemic average swim performance times. FR = freestyle, FL = butterfly, BK = backstroke, BR = breaststroke, IM = individual medley.

**Table 2 ijerph-19-00164-t002:** Top 16 male swimming performances in the GLIAC Regional Championship meets in a non-pandemic versus pandemic year. All swim events are reported in yards (yd) while all finishing times are reported in seconds.

Swim Event	Performance Times	Performance Times
	Non-Pandemic	Pandemic
	(2019–2020)	(2020–2021)
	Mean ± SD	Mean ± SD
	*(Fastest–Slowest)*	*(Fastest–Slowest)*
50 FR	20.8 ± 0.3	20.8 ± 0.5
	*(20.2–21.3)*	*(19.7–21.3)*
100 FR	45.7 ± 0.6	46.0 ± 0.9
	*(44.7–46.5)*	*(43.9–46.9)*
200 FR	101.1 ± 1.3	102.1 ± 1.5
	*(98.7–103.3)*	*(98.9–104.0)*
** *500 FR* **	** ** 276.1 ± 3.8* **	** *280.2 ± 5.4* **
	** *(271.4–281.5)* **	** *(272.1–289.1)* **
1000 FR	597.0 ± 97.2	577.2 ± 16.1
	*(550.4–959.8)*	*(540.5–594.6)*
1650 FR	964.4 ± 21.2	979.3 ± 33.4
	*(923.0–992.3)*	*(916.5–1034.1)*
100 FL	50.0 ± 0.7	50.2 ± 0.9
	*(47.8–50.7)*	*(48.0–51.3)*
200 FL	114.1 ± 2.1	114.0 ± 3.2
	*(109.8–116.6)*	*(108.7–117.8)*
100 BK	51.0 ± 1.4	50.9 ± 1.3
	*(48.2–53.6)*	*(48.1–52.8)*
200 BK	114.7 ± 6.2	112.7 ± 3.6
	*(106.8–128.2)*	*(105.9–117.5)*
100 BR	57.7 ± 1.8	57.1 ± 1.8
	*(54.9–61.0)*	*(54.1–59.9)*
200 BR	125.6 ± 2.9	126.7 ± 4.0
	*(120.2–130.0)*	*(120.4–132.7)*
200 IM	113.1 ± 2.2	113.5 ± 2.2
	*(108.9–115.5)*	*(109.2–115.9)*
400 IM	246.0 ± 5.5	247.6 ± 7.4
	*(237.2–253.9)*	*(235.2–258.0)*

* *p* < 0.05 between pandemic and non-pandemic average swim performance times. FR = freestyle, FL = butterfly, BK = backstroke, BR = breaststroke, IM = individual medley.

**Table 3 ijerph-19-00164-t003:** Top 16 female swimming performances in the NCAA D2 National Championship meets in a non-pandemic versus pandemic year. All swim events are reported in yards (yd) while all finishing times are reported in seconds.

Swim Event	Performance Times	Performance Times
	Non-Pandemic	Pandemic
	(2018–2019)	(2020–2021)
	Mean ± SD	Mean ± SD
	*(Fastest–Slowest)*	*(Fastest–Slowest)*
** *50 FR* **	** *** 23.1 ± 0.3* **	** *23.4 ± 0.2* **
	** *(22.4–23.4)* **	** *(22.8–23.6)* **
** *100 FR* **	** *** 50.2 ± 0.7* **	** *50.8 ± 0.5* **
	** *(48.4–50.8)* **	** *(50.0–51.4)* **
200 FR	109.5 ± 0.8	110.0 ± 0.8
	*(108.0–110.4)*	*(108.4–110.9)*
500 FR	294.6 ± 1.8	295.7 ± 2.3
	*(291.6–297.0)*	*(291.8–299.2)*
1000 FR	603.4 ± 8.1	608.2 ± 7.3
	*(585.9–613.7)*	*(593.2–618.6)*
1650 FR	1010.9 ± 14.4	1012.9 ± 14.1
	*(980.1–1023.5)*	*(991.0–1034.1)*
100 FL	54.5 ± 0.8	54.8 ± 0.8
	*(52.5–55.4)*	*(53.0–56.0)*
200 FL	121.5 ± 0.9	122.6 ± 2.1
	*(120.0–122.8)*	*(119.1–126.1)*
100 BK	54.6 ± 0.8	55.1 ± 0.9
	*(52.5–55.5)*	*(52.9–56.2)*
200 BK	119.2 ± 1.5	119.7 ± 1.3
	*(116.1–120.9)*	*(116.6–121.6)*
100 BR	62.2 ± 0.7	62.6 ± 0.4
	*(60.5–62.9)*	*(62.0–63.4)*
200 BR	135.9 ± 1.7	136.6 ± 1.8
	*(132.1–138.0)*	*(133.0–139.1)*
200 IM	121.7 ± 1.4	122.5 ± 1.8
	*(118.8–123.8)*	*(119.1–124.7)*
** *400 IM* **	** ** 262.6 ± 2.8* **	** *259.8 ± 3.7* **
	** *(255.1–265.6)* **	** *(254.2–264.7)* **

* *p* < 0.05; ** *p* < 0.01 between pandemic and non-pandemic average swim performance times. FR = freestyle, FL = butterfly, BK = backstroke, BR = breaststroke, IM = individual medley.

**Table 4 ijerph-19-00164-t004:** Top 16 male swimming performances in the NCAA D2 National Championship meets in a non-pandemic versus pandemic year. All swim events are reported in yards (yd) while all finishing times are reported in seconds.

Swim Event	Performance Times	Performance Times
	Non-Pandemic	Pandemic
	(2018–2019)	(2020–2021)
	Mean ± SD	Mean ± SD
	*(Fastest–Slowest)*	*(Fastest–Slowest)*
** *50 FR* **	** *** 20.0 ± 0.2* **	** *19.8 ± 0.2* **
	** *(19.6–20.3)* **	** *(19.3–20.0)* **
100 FR	43.8 ± 0.3	43.7 ± 0.4
	*(43.1–44.2)*	*(43.0–44.2)*
200 FR	96.8 ± 0.9	96.7 ± 1.5
	*(94.0–97.5)*	*(93.3–98.3)*
500 FR	266.3 ± 1.2	266.4 ± 2.1
	*(264.5–268.1)*	*(263.1–269.4)*
1000 FR	552.1 ± 6.7	553.6 ± 6.9
	*(536.8–560.6)*	*(534.1–561.0)*
1650 FR	927.0 ± 10.1	928.4 ± 12.1
	*(906.4–939.9)*	*(895.1–941.6)*
100 FL	47.4 ± 0.7	47.0 ± 0.7
	*(45.0–48.1)*	*(45.6–47.9)*
200 FL	107.0 ± 1.5	106.9 ± 1.6
	*(102.5–108.6)*	*(102.9–108.9)*
100 BK	47.5 ± 0.4	47.7 ± 0.4
	*(46.7–48.3)*	*(47.0–48.3)*
** *200 BK* **	** ** 105.2 ± 0.8* **	** *105.9 ± 0.7* **
	** *(104.0–106.6)* **	** *(104.5–107.0)* **
100 BR	54.2 ± 0.6	54.2 ± 0.7
	*(52.9–54.8)*	*(52.4–55.0)*
200 BR	118.8 ± 1.1	119.0 ± 1.3
	*(116.9–120.1)*	*(116.7–121.0)*
200 IM	107.6 ± 0.9	107.5 ± 1.1
	*(105.6–109.0)*	*(105.1–108.8)*
400 IM	232.8 ± 1.7	231.4 ± 2.7
	*(230.4–235.6)*	*(225.8–235.2)*

* *p* < 0.05; ** *p* < 0.01 between pandemic and non-pandemic average swim performance times. FR = freestyle, FL = butterfly, BK = backstroke, BR = breaststroke, IM = individual medley.

## Data Availability

All data will be available on request to the corresponding author (T.H.-B.).

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
