# Peer review of "Pandemic-Induced Reductions on Swim Training Volume and Performance in Collegiate Swimmers"

_ijerph, 2021, doi:10.3390/ijerph19010164_

Round 1

Reviewer 1 Report

Comments to the authors:

  1. One of the major concerns is that swimmers’ age is not addressed in the paper. Moreover, age and training age is also different. The paper covers a 3-year period, there is a declination of performance with aging and also there is an inclination with getting more training age. Mean age and years in training should be addressed in the Methods and Results sections.
  2. The second major issue is the Performance time comparisons in the Tables. What is the competition format? Do they have heats / finals? If yes, which time did you choose and why? Please describe the competition structure.

The meaning of these tables is questionable because they compare the top 16 male swimming performances in the GLIAC Regional Championship meets and the top 16 female swimming performances in the NCAA D2 National Championship meets, in a non-pandemic versus pandemic year. Why do you use two different Championships to compare the male and female performance? Regional Championships are usually lower quality than Nationals, especially the density of the top performances is more variable. The top 16 male and female swimmers’ performances were compared; it means that not the performance of the same swimmers. All swim events are reported in yards (yd) while all finishing times are reported in seconds. I suggest to use a score point system like FINA points or performance times compared to a swimmers’ personal best result. To improve a performance on a 50 m distance (milliseconds) is not comparable to an improvement on 400 m distance.

  1. In the COVID pandemic, dryland training was more pronounced in athletes’ life. Do you have any information about the dryland training hours before and after the lockdown? May it contribute to the results that Sprinters had more strengths training, or did they have endurance training on dryland, which you hoped for a cross training effect in the water (VO2 max protocols on treadmill or similar…)? Is there any data in the literature of this?
  2. How long was the resting period before the pandemic versus the non-pandemic year? It may contribute to the results, so it would be good to detail in the Methods. If you don’t have this data, you should address this limitation section.
  3. Only 8 swimmers were allowed to train in the pool at the same time. Were the training hours shorter with more intensity. Was there a complete lock-down with home training? What was the number of races in the pandemic versus non-pandemic years? What did they do between 2020 March to September? What is the training amount (hours) that they train in a week? I would complement the Methods with this information. Could you show Figure 1 with a monthly data about training volumes? The exact number of races is important. Is there a relationship between the amount of races swam and the development of the swimmers? Like: if one swimmer had a lot of races (more individuals, relays…), did she/he develop differently, than somebody with few races?
  4. A Figure would be very helpful to better understand the periodization and training phases in the pandemic versus non-pandemic years.
  5. The number of COVID infections were low as in other studies. Were there other medical issues which consequence training stop? How many swimmers were not infected, but needed to stop training because they had to quarantine (close contacts)? Was there any kind of return to play protocol for close contacts?
  6. Could you add any objective performance parameter such as heart rate, lactate, BORG or Foster score that would verify better or lower performance (training intensity)?

Reviewer 2 Report

I would like to thank you for submitting and give me the opportunity to review the manuscript entitled: “Pandemic-induced Reductions on Swim Training Volume and Performance in Collegiate Swimmers”. The research topic undertaken by the authors is very interesting in times of pandemic due to the limitations and changes to which competitive athletes are undergoing. These results may be of great importance for increasing the knowledge of the influence and importance of the mental and physical conditions of the athletes during training and competitions. The study is well performed, the results are compelling and adequately presented. Nevertheless, some questions and concerns need to be answered and corrected before the formal acceptance of the manuscript. In this sense, I only have a few minor comments.

Regarding the first objective, in which it is determined how pandemic-induced training restrictions affected training volume, such as swimming distance, in a NCAA Division swim team compared with a non-pandemic year, the local swimming team was divided into 3 groups according to training group regardless of gender. Why don't the authors study the difference according to gender as they do in the other objectives?

Although it appears in results, in materials and methods the sample size of the first objective is absent. It is necessary to specify.

In the tables of the results section, for a quicker and more comprehensive visualization, I suggest marking in bold the data with significant p values.

In the discussion, while in one section it states that the drop-out rate was equal between sexes which conflicts with pre-pandemic findings (mental health symptoms are typically higher in female versus male elite athletes) (lines 269-270), in another section it is specified that females, at lower levels of competition, were collectively more susceptible to mental health challenges (line 308). These two concepts, although supported by other scientific studies, are opposed to each other and a better argumentation is suggested. In the same sense, it is suggested to hypothesize about some of the reasons for why the authors think that the performance effects from the COVID-19 lockdown are higher at the local level than at the elite national level.

Finally, it is recommended to check some aspects such as the spelling, since, for example, “championship” sometimes starting with capital letter and sometimes not. Delete "for reasons" in line 318 as it is duplicated. Least on line 330 is in italics. Check the year of publication of the reference 3.
